# MicroRNAs in the Regulation of Endometrial Receptivity for Embryo Implantation

**DOI:** 10.3390/ijms23116210

**Published:** 2022-06-01

**Authors:** Manizha Shekibi, Sophea Heng, Guiying Nie

**Affiliations:** Implantation and Pregnancy Research Laboratory, School of Health and Biomedical Sciences, Royal Melbourne Institute of Technology, Bundoora, VIC 3083, Australia; s3862633@student.rmit.edu.au (M.S.); sophea.heng@rmit.edu.au (S.H.)

**Keywords:** microRNA, miRNA, endometrium, endometrial receptivity, embryo implantation, IVF

## Abstract

Development of endometrial receptivity is crucial for successful embryo implantation and pregnancy initiation. Understanding the molecular regulation underpinning endometrial transformation to a receptive state is key to improving implantation rates in fertility treatments such as IVF. With microRNAs (miRNAs) increasingly recognized as important gene regulators, recent studies have investigated the role of miRNAs in the endometrium. Studies on miRNAs in endometrial disorders such as endometriosis and endometrial cancer have been reviewed previously. In this minireview, we aim to provide an up-to-date knowledge of miRNAs in the regulation of endometrial receptivity. Since endometrial remodelling differs considerably between species, we firstly summarised the key events of the endometrial cycle in humans and mice and then reviewed the miRNAs identified so far in these two species with likely functional significance in receptivity establishment. To date, 29 miRNAs have been reported in humans and 15 miRNAs in mice within various compartments of the endometrium that may potentially modulate receptivity; miRNAs regulating the Wnt signalling and those from the let-7, miR-23, miR-30, miR-200 and miR-183 families are found in both species. Future studies are warranted to investigate miRNAs as biomarkers and/or therapeutic targets to detect/improve endometrial receptivity in human fertility treatment.

## 1. Introduction

Embryo implantation requires a well-developed embryo implanting into the inner lining of the uterus called the endometrium. For implantation to succeed, the endometrium must undergo major molecular and cellular changes beforehand to transform from a nonreceptive to a receptive state. A blastocyst cannot implant if the endometrium is not ‘receptive’ [1]. The human endometrium is a dynamic tissue that undergoes cyclic remodelling across the menstrual cycle, becoming receptive for embryo implantation only for a short period of time in each cycle [1,2]. It is estimated that approximately one third of implantation failures in healthy women are likely due to inadequate endometrial receptivity [3]. Implantation failure is also commonly experienced by those who undergo fertility treatment such as *in vitro fertilization* (IVF) where ~60–70% of high-quality embryos fail to implant, and non- or sub-receptive endometrium is believed to be an important contributor [4,5]. Understanding the molecular regulation of endometrial receptivity is thus critical to improve fertility in general and to increase implantation rates in fertility treatment.

Endometrial remodelling differs between humans (undergoing the menstrual cycle) and animal species such as mice (undergoing the oestrous cycle) [6,7]. It is also known that the development of receptivity involves multifaceted changes within various endometrial cell types and the intrauterine environment [8,9,10]. This minireview firstly synopsizes the endometrial cycle in humans and mice. It then discusses microRNAs (miRNAs) that have been discovered in different compartments of the endometrium during the development of receptivity in both species. MiRNAs secreted from the endometrium and/or embryos that may influence endometrial receptivity are also briefly reviewed. Discussions are then made on miRNAs as potential biomarkers or therapeutic targets in human fertility treatment. 

## 2. Brief Summary of the Endometrial Cycle in Humans and Mice

### 2.1. The Human Menstrual Cycle and Endometrial Receptivity

The human endometrium consists of two distinctive regions, the basalis and the functionalis [11,12] (Figure 1A). The basalis is a permanent layer that acts as the base for endometrial regeneration following menstruation [7], whereas the functionalis thickens and remodels during the menstrual cycle; it is this part of the endometrium that facilitates embryo implantation in a conceptual cycle but sheds in every nonconceptual cycle [1]. The menstrual cycle is regulated primarily by ovarian steroid hormones, especially estrogen and progesterone [7], and endometrial changes across an average 28-day cycle can be categorized into three phases (Figure 1B): menstrual, proliferative and secretory [7]. The menstrual phase involves the shedding of the functionalis following the rapid decline in estrogen and progesterone due to degeneration of the corpus luteum (CL) in the ovary. Menses eliminates the decidualised tissue which is irreversibly differentiated; the endometrium then regrows and reprepares for potential pregnancies in the subsequent cycle [13].

The proliferative phase is the regeneration of the functionalis. Major activities include re-epithelialisation while maintaining developmental plasticity to prevent endometrial scarring [14,15] and elongation of glands and vasculature to form a mature endometrium [14,16]. Sufficient tissue growth during this phase is crucial for the development of receptivity in the next phase, and estrogen produced by the developing ovarian follicles is the dominant hormone [7]. Estrogen receptors α (ER-α) and β (ER-β) are both expressed in the endometrium and peak during the proliferative phase [7,17], with ER-α in epithelial/stromal cells and blood vessels and ER-β predominantly in the epithelium [7,17]. 

Following ovulation, the thickened endometrial tissue enters the secretory phase, starting to differentiate in anticipation of embryo implantation (Figure 1B). Progesterone produced by the CL peaks in the mid-secretory phase and acts as the key driver of multiple molecular and morphological changes for the development of endometrial receptivity [2,18,19]. For instance, endometrial epithelial cells become highly secretory, while the stroma undergoes a terminal differentiation process known as decidualisation [1,20]. Decidualised cells secrete the necessary extracellular matrix required for a successful pregnancy [1,13,20]. Both ER-α and ER-β are downregulated by progesterone in the secretory phase [7,14,17].

The endometrium becomes transiently receptive during the mid-secretory phase (day 20–24), approximately 6–10 days after the ovulation (Figure 1B) [2,21], and this period is also known as the “window of implantation” [21,22]. During implantation (Figure 1C), the blastocyst firstly apposes itself to the surface of the receptive endometrium, establishing a loose connection between the trophectoderm of the blastocyst and the luminal epithelium of the endometrium [23,24]. Thereafter, the blastocyst firmly attaches to the luminal epithelium and then invades this layer of epithelial cells and moves to the stromal compartment. The luminal epithelium then reseals and encloses the embryo within the endometrial tissue [25]. 

Embryo transfer outside of the receptive window results in implantation failure or increased risk of early pregnancy loss [3,26], highlighting the importance of endometrial receptivity. To date, many molecular changes associated with endometrial receptivity have been found and have been reviewed previously [9,27,28,29,30,31,32]. Because the blastocyst directly interacts with the luminal epithelium to initiate implantation, changes in this layer of endometrial epithelial cells are essential for endometrial receptivity [23,24]. Past studies have shown that luminal epithelial cells are well-polarised during the nonreceptive state, but polarity-supporting factors are downregulated at receptivity [32]. Progesterone is known to upregulate adhesion molecules during the secretory phase [33]. Recently, we have discovered membrane protein podocalyxin (PODXL) as the key negative regulator of human endometrial epithelial receptivity [34]. PODXL is expressed in all epithelial (luminal and glandular) and endothelial cells in the nonreceptive phase but downregulated selectively in the luminal epithelium in the receptive phase [34]. Functionally, PODXL inhibits embryo attachment and invasion in in vitro models [34,35], and insufficient downregulation of PODXL in the luminal epithelium is associated with implantation failure in patients undergoing IVF treatment [35]. At the molecular level, PODXL promotes an impermeable/polarized epithelium and inhibits pro-implantation factors to negatively control endometrial epithelial receptivity [36]. PODXL is thus a potentially important regulator of human endometrial epithelial receptivity. 

### 2.2. The Murine Estrous Cycle and the Key Differences in Endometrial Receptivity Development between Mice and Humans 

The murine uterus, consisting of two horns, undergoes the oestrous cycle (Figure 2). Each cycle lasts approximately 4–5 days and comprises four phases: proestrus, oestrus, metestrus and diestrus [14,37]. Proestrus is characterised by increased blood flow and distension of the uterus, while epithelial cells and leukocytes begin to proliferate. At oestrus, leukocyte numbers drop significantly and epithelial cell proliferation increases. During metestrus, epithelial cells begin to degenerate, leukocytes proliferate once again and the uterus is no longer distended. The degenerated tissue is then reabsorbed by the uterus during diestrus [6,38,39]. Thus, the uterine tissue does not shed, and no bleeding occurs during the oestrous cycle.

The murine oestrous cycle is also controlled by ovarian hormones estrogen and progesterone. However, unlike humans in whom endometrial receptivity is established in the mid-secretory phase of the menstrual cycle, female mice are sexually active only during the oestrous phase, and the endometrium becomes receptive around 4.5 days post-coitus [38,40]. Among many molecular changes, a transient rise of oestradiol on day 4 of pregnancy, the nidatory oestradiol surge, is key to triggering the expression of several endometrial factors, such as the leukemia inhibitory factor (LIF), to fully induce endometrial receptivity for implantation [41]. 

Furthermore, while shifts in hormonal levels during the menstrual cycle induce spontaneous endometrial stromal decidualisation in humans [13], the murine uterus does not undergo decidualisation until after an embryo has contacted the uterine epithelium for implantation [42]. Additionally, the orientation of the blastocyst during implantation differs between humans and mice. In humans, the polar trophectoderm (located nearest to the inner cell mass, ICM) of the blastocyst interacts with the endometrial luminal epithelium (Figure 1C), but in mice, the mural trophectoderm (located farthest from the ICM) interacts with the endometrial luminal epithelium at implantation (Figure 2) [38]. In humans, the blastocyst invades between luminal epithelial cells, whereas in mice, luminal epithelial cells undergo apoptosis at the site of embryo attachment to make space for the implanting blastocyst [43,44]. 

Due to these differences, whilst some characteristics identified in mice can be translatable to humans, mice are often not the most appropriate animal model to study human endometrial receptivity, especially epithelial receptivity, for embryo implantation. 

## 3. MicroRNAs and Endometrial Receptivity 

MiRNAs are an important class of non-coding RNAs; they are endogenous and single-stranded regulatory RNAs which can vary between 18 and 25 nucleotides in length [45,46,47]. MiRNAs function as regulators of mRNA and mainly target the 3′ untranslated region (UTR) of gene transcripts [48,49]. They bind to Argonaute proteins to form RNA-induced silencing complexes and cause cleavage or translational repression of mRNA targets [50]; the degree of sequence complementation between the miRNA and the mRNA target dictates the strength of regulation [47]. Currently, over 2500 mature miRNAs have been identified [48], many of which are shown to be involved in reproduction and pregnancy. 

Previous reviews discussed miRNAs in reproduction, covering both the male and female systems [51,52,53,54]; those on pregnancy placed more emphasis on miRNAs and maternal immune tolerance [55,56] or pregnancy complications [57,58,59]. MiRNAs in gynaecological diseases have also been reviewed [60,61,62]. However, the endometrium has often been overlooked in reviews concerning miRNAs, except when endometrial miRNAs are related to endometriosis [63,64,65,66,67,68,69,70] or endometrial cancer [71,72,73,74,75,76]. The remainder of this paper thus focuses on miRNAs in endometrial receptivity which has not been reviewed previously. Studies in both humans and mice are discussed.

### 3.1. MiRNAs Identified in the Endometrial Tissue

Not all studies of miRNAs in endometrial receptivity focused on a particular cell type; instead, many analysed endometrial tissues without revealing the exact location of miRNA production/action. The key miRNAs identified by this approach are listed in Table 1. 

Two separate studies identified the LIF in the murine environment as a target gene of miRNAs, specifically of miR-181 [77] and miR-223 [78]. LIF has been well-established to play an important role in endometrial receptivity and implantation [79]. LIF is shown to be upregulated in fertile women compared to infertile women during the mid-secretory phase, suggesting its importance for receptivity [80]. In the studies assessing miRNAs that target LIF, upregulation of miR-181 and miR-223 supressed LIF expression in the murine endometrium and compromised embryo implantation [77,78]. Potential miRNA regulation of LIF was also identified in humans, specifically by miR-494 [81]. In silico prediction tools identified LIF as a target of miR-494 in the human endometrial tissue, where downregulation of miR-494 was associated with a high expression of LIF [81]. LIF appears to be a common target of multiple miRNAs in the development of endometrial receptivity; however, it is unknown whether some of these miRNAs are more physiologically important than others.

Another in silico study identified a further nineteen miRNAs that are potentially involved in receptivity development [3], in particular miR-130b, miR-548n/ah, miR-30c and miR-449, which are downregulated during the receptive phase [3]. It is of note that downregulation of miR-30c has also been associated with endometrial cancer [82]. It has been predicted that downregulation of these miRNAs during the receptive phase corresponds to the upregulation of 11 target genes, which include ephrin-A1 (EFNA1), growth arrest and DNA damage-inducible protein 45 alpha (GADD45A), annexin A4 (ANXA4), secreted phosphoprotein 1 (SPP1), AT-rich interaction domain 5B (ARID5B), interleukin 15 (IL15), Dickkopf-related protein 1 (DKK1), insulin-like growth factor binding protein 1 (IGFBP1) [3].

Other miRNAs found to regulate receptivity in the murine endometrial tissue include miR-101 and miR-199a, which target the cyclooxygenase-2 gene (Cox-2) [83]. Cox-2 was previously identified as a critical molecule for murine implantation and appears to have an inverse relationship with miR-101 and miR-199a [83]. However, another study reported that miR-199a may also target mucin-1 (MUC1), which has been established to maintain murine endometrial epithelial cells in a non-receptive state [84]. It is also reported that miR-193, which targets growth factor receptor-bound protein 7 (*GRB7*) [85], is involved in murine embryo implantation. However, the exact role of *GRB7* in endometrial receptivity is unknown. Additionally, studies using human endometrial tissue identified miR-135b to be elevated in infertile women, and this miRNA is shown to downregulate homeobox A10 (HOXA10) expression [86]. 

A recent study profiled murine uterine miRNA transcriptomes at 1 day (non-receptive), 4 days (receptive) and 5 days (implantation) post-coitus [87]; 138 miRNAs were found to be differentially expressed across these timepoints, and the five most upregulated (miRNAs, miR-27a, miR-96, miR-106a, mir-23a and miR-200b) targeted a substantial proportion of the detected mRNAs [87]. Among these five miRNAs, miR-27a has also been detected in human serum samples; however, miR-27a is upregulated in patients with chronic endometritis [88] with no association with pregnancy outcomes. 

**Table 1 ijms-23-06210-t001:** The miRNAs identified in endometrial tissues.

miRNA	Species	Target Gene/Protein	Improved/Impaired Receptivity	Ref.
miR-30c	Human	↓EFNA1, ARID5B	Impaired	[3]
miR-101	Mouse	↓*Cox-2*	Impaired	[83]
miR-130b	Human	↓ANXA4, IL15	Impaired	[3]
miR-135b	Human	↓HOXA10	Impaired	[86]
miR-181	Mouse	↓LIF	Impaired	[77]
miR-193	Mouse	↓*GRB7*	Impaired	[85]
miR-199a	Mouse	↓*Cox-2,* MUC1	Variable between gene targets	[83,84]
miR-223	Mouse	↓LIF	Impaired	[78]
miR-449c	Human	↓DKK1, IGFBP1	Impaired	[3]
miR-548ah/n	Human	↓EFNA1, GADD45A, ANXA4, SPP1, ARID5B	Impaired	[3]

### 3.2. MiRNAs Identified in Endometrial Epithelial Cells

MiRNAs identified to promote endometrial epithelial receptivity include the let-7 family [89,90], miR-23b [91], miR-30b/d [81,92], miR-183-5p [93], miR-192 [94] and miR-494 [81] (Table 2). On the other hand, miR-23a-3p [95], miR-30a-3p [96], miR-30d [97], miR-125b [98], miR-145 [99,100], miR-200c [101], miR-429 and miR-5088 [102] are reported to hinder epithelial receptivity (Table 2).

While both the let-7-a and let-7-g families of miRNAs affect murine endometrial receptivity, only let-7-a is involved in human endometrial receptivity [89]. This particular study identified that Wnt signalling impairs uterine receptivity and that let-7-a/g, via downregulating β-catenin, represses Wnt to improve endometrial receptivity [89]. MiR-23b is also shown to indirectly influence Wnt signalling in women with recurring implantation failure (RIF) via targeting secreted frizzled-related protein 4 (sFRP-4), a modulatory protein of the Wnt pathway. Women with RIF are shown to have elevated levels of miR-23b, causing significantly lower levels of sFRP-4 in comparison to fertile women [91]. As sFRP-4 negatively regulates the Wnt pathway to increase endometrial receptivity, downregulation of sFRP-4 in RIF patients potentially impairs their receptivity [91]. Another miRNA, miR-183-5p, which targets catenin alpha 2 (CTNNA2), is reported to indirectly regulate the Wnt signalling pathway in the both murine and human endometrial epithelial cell line Ishikawa [93]. Both sFRP-4 and CTNNA2 negatively regulate the Wnt pathway to increase endometrial receptivity [91,93].

However, one study suggests that miR-23a downregulates cullin-3 (CUL3) to promote receptivity in Ishikawa cells [95], whereas other studies report that downregulation of CUL3 causes upregulation of β-catenin, ultimately impairing rather than increasing endometrial receptivity and implantation [89,91,93]. Therefore, there are conflicting results regarding the role of miR-23a in the regulation of endometrial receptivity.

Besides β-catenin, MUC1 has also been identified as a target of let-7-a/g in mice [90]. Downregulation of MUC1 by let-7 improves, whereas upregulation of MUC1 by miR-192 impairs endometrial receptivity in mice [94]. This indicates that in the murine endometrium, regulation of MUC1 via let-7, miR-192 and other miRNAs that are yet to be identified would modulate receptivity. MiR-192 also upregulates the adherens junction protein E-cadherin, which may also contribute to the impairment of endometrial receptivity in mice [94]. Furthermore, miR-192 downregulates Rho GTPase-activating protein 19 (ARHGAP19), which is involved in the remodelling of junctional proteins in murine endometrial epithelial cells [103]. 

The miR-30 family has been investigated in endometrial epithelial receptivity. Two extensive microarray and proteomics studies using receptive phase human endometrial epithelial cells have identified hundreds of gene targets of miR-30b/d, with varying degrees of up- or downregulation [81,92]. In mice, upregulation of miR-30a-3p and miR-30d has opposing impacts: elevated miR-30a-3p reduces receptivity by suppressing snail family transcriptional repressor 2 (*Snai2*) [96], whereas miR-30d deficiency impairs overall implantation [97]. 

In the previously mentioned study by Altmäe et al. [81]*,* miR-494 also has hundreds of potential targets, but the following are particularly of note: calpastatin (CAST), cystic fibrosis transmembrane conductance regulator (CFTR), dihydropyrimidinase-like 2 (DPYSL2), F11 receptor (F11R), fibroblast growth factor receptor 2 (FGFR2), LIF, metal-regulatory transcription factor 1 (MTF1), neuronal PAS domain protein 2 (NPAS2), peroxisome proliferator-activated receptor gamma coactivator 1 alpha (PPARGC1A), transforming acidic coiled-coil-containing protein 2 (TACC2) and RAB40B. While all display some degree of regulation by miR-494, the downregulation of miR-494 in the receptive phase leads to 5–10× greater upregulation of the CFTR and LIF than of the other target genes [81]. 

Multiple studies of RIF have identified miR-145 as an important regulator of endometrial receptivity [91,99,100,102]. All these studies have used human endometrial tissue, and studies by Liu et al. [100] and Kang et al. [99] also used the Ishikawa cell line to identify cell-specific roles of miR-145. This miRNA has been found to primarily play a role in cell–cell interaction and junctional proteins [91,99] and regulate ER-α [102] and plasminogen activator inhibitor (PAI-1) [100]. In the study by Shi et al. [102], miR-145 is lower in patients with RIF; the same study also identified miR-429 and miR-5088 being upregulated and miR-4668-5p downregulated in patients with RIF [102].

Furthermore, miR-125b [98] and miR-200c [101] are reported to impair epithelial receptivity in the murine epithelium. MiR-125b represses the expression of matrix metallopeptidase 26 (MMP26), a major player of endometrial remodelling, to reduce endometrial receptivity [98]. On the other hand, miR-200c influences endometrial receptivity via indirectly targeting the Wnt pathway [101]. Through targeting the gene fucosyltransferase 4 (FUT4), miR-200c is suggested to indirectly inactivate β-catenin/Wnt signalling, leading to impaired endometrial receptivity and implantation [101]. The β-catenin/Wnt signalling pathway is emerging to be important in the regulation of endometrial epithelial receptivity. However, whether the pathway helps or hinders endometrial receptivity appears to be in question as the related molecular mechanisms are not yet fully understood.

### 3.3. MiRNAs Identified in Endometrial Stromal Cells 

While endometrial epithelial cells are the first point of contact by the blastocyst to initiate implantation, endometrial stromal cells are vital in subsequent stages of implantation. As decidualisation must occur for implantation to progress properly, studies have also investigated the role of miRNAs in decidualisation in humans and mice. MiRNAs that have been identified as the key players in promoting decidualisation include miR-21 [104], miR-96 [87,105], miR-181a [106] and miR-200 [107] (Table 3); miRNAs associated with impairment of decidualisation consist of miR-22 [108], miR-141 [109], miR-148a [110], miR-181b [111], miR-194 [112], miR-200a [113,114] and miR-542 [115] (Table 3). 

Using primary stromal cells isolated from the human endometrium, the studies by Yan et al. [104] and Zhang et al. [106] both identified Kruppel-like factor 12 (KLF12) as the target gene of miR-21 and miR-181a. Both miRNAs downregulate KLF12 to promote decidualisation in vitro. KLF12 was previously identified as a negative regulator of decidualisation in patients with RIF [104], and overexpression of KLF12 is associated with downregulation of decidualisation markers [106]. MiR-200 is also reported to be associated with decidualisation in human endometrial stromal cells via repressing zinc finger E-box binding homeobox 1 (ZEB1) expression [107]. 

In murine endometrial stromal cells, miR-96 is shown to play an important role in decidualisation [87,105]. The study by Yang et al. [105] identified miR-96 levels inversely correlating with the expression of antiapoptotic protein B-cell lymphoma 2 (Bcl2). In the study of Chen et al. [87], miR-96 targets Kruppel-like factor 13 (*Klf13*) and potentially regulates the progesterone receptor (*PGR*) in mice. This indicates that miR-96 likely plays a vital role in the decidualisation of murine endometrial stromal cells. 

However, the expression of certain miRNAs can cause dysregulation of decidualisation. Patients with RIF are reported to have elevated levels of miR-22 with reduced levels of *Tiam/Rac1*, a signal pathway considered to be potentially important in early stromal cell decidualisation [108]. Overexpression of miR-148a is also associated with impairment of decidualisation in patients with RIF via the suppression of homeobox C8 (HOXC8) [110]. Furthermore, miR-542 [115] and miR-194 [112] are reported to dysregulate decidualisation of human endometrial stromal cells. In a human endometrial stromal cell line, miR-181b downregulates tissue inhibitor of metalloproteinase 3 (TIMP-3), but overall does not significantly affect decidualisation [111]. 

Two different miRNAs have been found to target the same gene in murine endometrial stromal cells, where upregulation of miR-141 [109] and miR-200a [113] negatively regulate the phosphatase and tensin homolog (PTEN) expression. Like Bcl2, PTEN is also believed to control proliferation and apoptosis of stromal cells, thus maintaining the correct environment for decidualisation [113]. Upregulation of miR-200a has also been associated with indirect downregulation of progesterone signalling in mice, reducing the overall implantation capacity [114]. 

### 3.4. MiRNAs Secreted by the Endometrium and/or the Embryo That May Influence Endometrial Receptivity

The endometrium can also be influenced by the implanting embryo. Several studies identified miRNAs that are expressed by embryos and present in embryo secretions [116,117,118,119,120]. Due to the increasingly common use of IVF techniques, blastocyst spent media have been analysed for miRNA and other biomarkers [116,117,118,119,120]. A few previous reviews discussed miRNAs and embryos, often regarding embryo development but also implantation [116,117,118,119,120]. Other previous reviews deliberated the miRNAs secreted by preimplantation blastocysts in facilitating the embryo–endometrium crosstalk to promote implantation [118,120]. Hence, no extensive discussion on this topic is made here. Embryo-derived miRNAs have also been suggested to play a role in implantation failure. For instance, miR-661 is identified to be secreted by human blastocysts, but uptake of this miRNA by human epithelial endometrial cells (HEECs) reduces their ability for trophoblast spheroids attachment in in vitro implantation models [121]. Other studies have identified miR-519d [122] and miR-155 [123] as negative regulators of trophoblast proliferation and migration during implantation. However, it is unknown if miR-519d and miR-155 also regulate endometrial cells for implantation [122,123]. 

While mRNA relies on the modifications to the RNA strand to improve its stability [124], miRNAs are only 18–25 nucleotides long [45,46,47] and generally do not undergo these modifications. Some miRNAs can circulate freely within the body, but the vast majority are located within extracellular vesicles (EVs), either in the form of microvesicles or exosomes [125,126], and circulate through the body while ‘packaged’. Multiple reviews have discussed EVs that are secreted from the endometrium and/or embryos and contain miRNAs, some of which are found to potentially enhance endometrial receptivity and embryo implantation [126,127,128,129]. Not all of the studies used human samples, but of those that did, the miR-30 family was identified multiple times as important miRNAs in human endometrial EVs [127,129]. 

The endometrium is proposed to have a mechanism to screen embryo quality prior to implantation [130], and EVs secreted by embryos have been suggested as the mediator where the cargo in the EVs is exposed to and assessed by the endometrium for embryo quality [130]. Embryonic EVs have been suggested to epigenetically regulate endometrial cells via EV-mediated RNA transfer, and poor embryos are unable to establish the same epigenetic changes [131]. There is growing evidence suggesting that the endometrium also has an active role in selecting embryonic EVs based on the cargo profile, which could include miRNAs such as the let-7 family, miR-31, miR-101a, miR-199a, miR-200 and the aforementioned miR-661 [121,132]. Other studies have predicted that miRNAs derived from the blastocyst culture medium may potentially modulate endometrial epithelial proliferation and embryo attachment [31,130,133]. 

## 4. Potential Clinical Applications

Since the first successful IVF birth in 1978 [134], more than 8 million babies have been born using assisted reproductive technologies, with the technique becoming commonplace globally [135,136]. However, as discussed earlier, implantation failure limits IVF treatment, and inadequate endometrial receptivity is an important contributing factor. With the identification of miRNAs as important players in the establishment of endometrial receptivity, the presence of miRNAs in serum may allow for clinical diagnosis of fertility and receptivity [116,117,118]. Furthermore, as the use of miRNAs as therapeutics has been pushed into clinical trials in other fields [137,138], the utility of miRNAs in endometrial preparation in IVF can also be speculated.

### 4.1. MiRNAs for Detecting/Predicting Endometrial Receptivity

While many studies have sought to identify miRNAs in specific cell populations within the endometrium or in relevant bodily fluids, a few studies have investigated miRNAs as clinical biomarkers for infertility and receptivity (Table 4). Some of these studies have compared miRNA in the endometrium and in serum to identify physiologically relevant biomarkers. These include miR-27a [88], miR-31 [139], mir-152 and miR-155 [140], which are found to be expressed in the endometrium and detected in serum. In the study by Kresowik et al. [139], miR-31 is elevated in both the endometrium and serum of normal fertile women in the window of implantation. Upregulation of miR-31 is associated with the downregulation of FOXP3 and CXCL12, which may be the potential target genes [139]. On the other hand, miR-27a, miR-152 and miR-155 have been shown to be upregulated in the endometrium and serum of patients with RIF [140] or chronic endometritis [88]. Studies by Ghaebi et al. [141] also identified miR-155 as a potential biomarker. However, miR-155 is reported to be downregulated in patients with RIF, contradicting the findings by Drissennek et al. [140]. Other miRNAs which are significantly altered in the serum of patients with RIF include miR-25, miR-93, miR-106b (upregulated) and miR-146a (downregulated) [141]. 

### 4.2. MiRNAs for Improving Endometrial Receptivity

Currently, most miRNA therapeutics trials are likely to be miRNA inhibitors over miRNA mimics [142]. However, patent applications for both types do exist and tend to be cancer-focused [142,143]. While not yet available in the market, some miRNAs have been in clinical trials, with anti-miRNA drug Miravirsen targeting miR-122 for the treatment of the hepatitis C virus [137]. While there is some interest in developing treatments to target the endometrium [144], no miRNA therapeutics are currently in clinical trials. While technologies required for effectively transferring miRNAs are still being heavily researched, the vast majority relies on the use of nanoparticles to prevent degradation of miRNAs or miRNA inhibitors [144,145,146,147]. Previous studies showed that endogenous EVs do exist in the uterine space and can safely deliver miRNAs between tissues [126,128,129]. The use of synthetically produced exogenous EVs may allow transfer of miRNA therapeutics to the endometrium and avoid unnecessary off-target sites. However, this is still an emerging field, and the types of nanoparticles and their surface molecules differ between studies [144,145,146,147].

## 5. Conclusions

The endometrium is a complex and dynamic tissue; understanding the molecular changes that govern endometrial receptivity is vital to improve fertility treatment. This review summarised miRNAs that have been identified to date as potential regulators of endometrial receptivity in both mice and humans. Because of the key species-specific differences in the fundamental biology of endometrial remodelling, it is important to bear in mind that discoveries made in mice are not automatically applicable to humans. Encouragingly, a few miRNAs have been found to potentially regulate endometrial receptivity in both species; these include those that influence the Wnt signalling pathway and those from the let-7, miR-23, miR-30, miR-200 and miR-183 families. Future studies need to further investigate the clinical significance of these miRNAs.

## Figures and Tables

**Figure 1 ijms-23-06210-f001:**
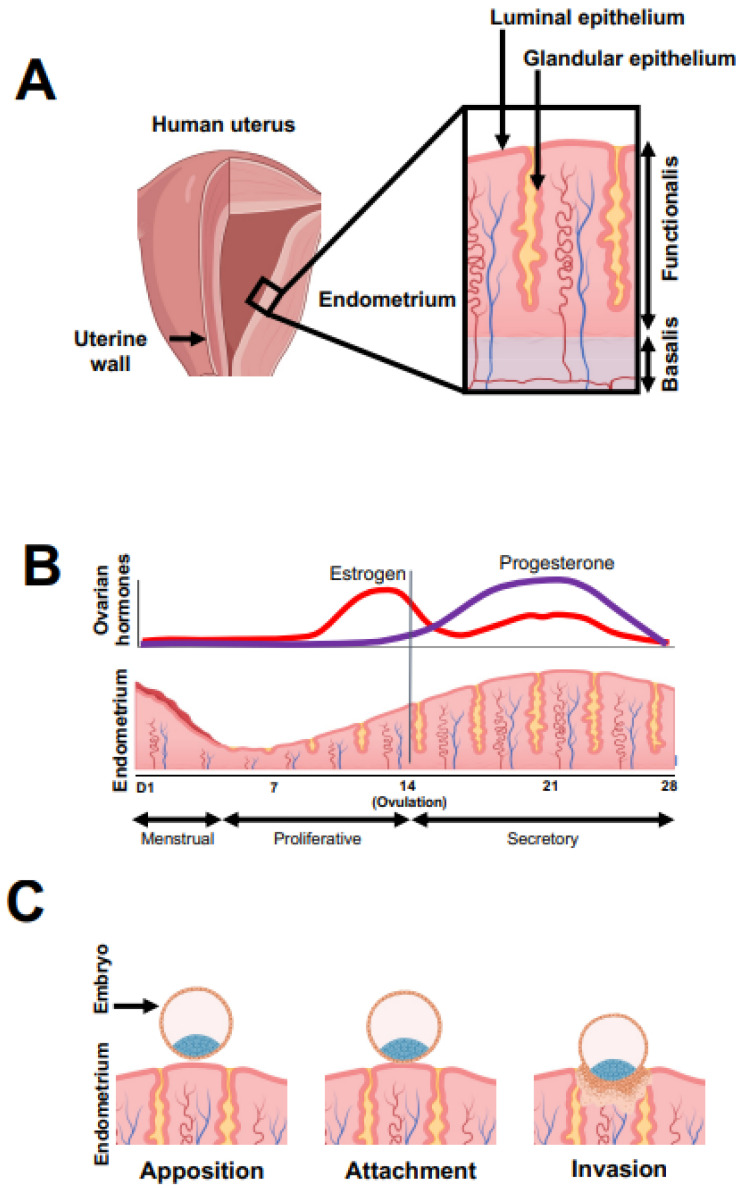
Schematic illustration of the human endometrium, its changes across the menstrual cycle and the initial process of embryo implantation. (**A**) The endometrium, the inner lining of the uterus, consists of the basalis and the functionalis and contains multiple cell types including luminal and glandular epithelial cells. (**B**) Endometrial changes across the menstrual cycle. The menstrual phase, lasting 1–5 days depending on the individual, marks the start of the cycle. The endometrium regrows in the proliferative phase and differentiates in the secretory phase. The fall in progesterone triggers the initiation of menses and the next cycle. (**C**) The apposition, attachment and invasion steps of embryo implantation. The embryo loosely apposes and then firmly attaches to the endometrial luminal epithelium; it then invades the stroma. Created with BioRender.com.

**Figure 2 ijms-23-06210-f002:**
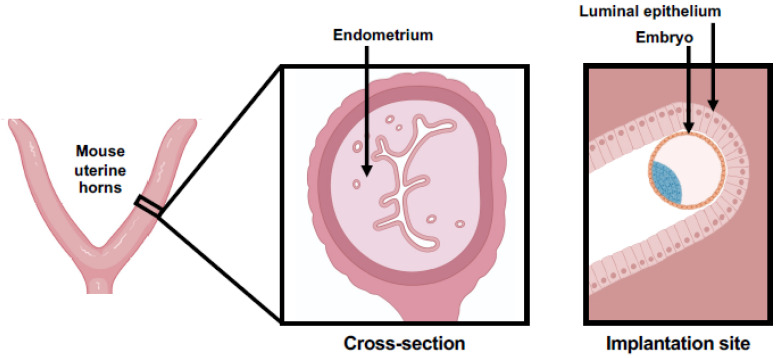
Schematic illustration of the murine uterus and the close contact between the blastocyst and the endometrium at the start of implantation. Shown are the two horns and a cross-section of the uterus, as well as the implantation site where an embryo is in the implantation chamber on pregnancy d4.5. Created with BioRender.com.

**Table 2 ijms-23-06210-t002:** The miRNAs identified in endometrial epithelial cells.

miRNA	Species	Target Gene/Protein	Improved/Impaired Receptivity	Ref.
let-7a/g	Human, mouse	↓β-catenin, MUC1	Improved	[89,90]
miR-23a/b	Human, mouse	↓CUL3, ↓sFRP-4	Variable between subtypes	[91,95]
miR-30a/b/d	Human, mouse	↓*Snai2*, ↓P4HA4	Variable between subtypes	[81,92,96,97]
miR-125b	Mouse	↓MMP26	Impaired	[98]
miR-145	Human	↓N-cadherin, IGF1R H2AFX, Netrin-4, ER-α, PAI-1	Impaired	[91,99,100,102]
miR-183	Human, mouse	↓CTNNA2	Improved	[93]
miR-192	Mouse	↑E-cadherin, Muc1, ↓ARHGAP19	Impaired	[94,103]
miR-200c	Mouse	↓FUT4	Impaired	[101]
miR-429	Human	↓DPP4, SERPING1, AQP3	Impaired	[102]
miR-494	Human	↓CAST, CFTR, DPYSL2, F11R, FGFR2, LIF, MTF1, NPAS2, PPARGC1A, TACC2, RAB40B	Impaired	[81]
miR-4668	Human	n/a	Improved	[102]
miR-5088	Human	↓DPP4, SERPING1, AQP3	Impaired	[102]

**Table 3 ijms-23-06210-t003:** The miRNAs identified in endometrial stromal cells.

miRNA	Species	Target Gene/Protein	Improved/Impaired Receptivity	Ref.
miR-21	Human	↓KLF12	Improved	[104]
miR-22	Human	↓*Tiam/Rac1*	Impaired	[108]
miR-96	Mouse	↓Bcl2, *Klf13*	Improved	[87,105]
miR-141	Mouse	↓PTEN	Impaired	[109]
miR-148a	Human	↓HOXC8	Impaired	[110]
miR-181a/b	Human	↓KLF12, TIMP-3	Variable between subtypes	[106,111]
miR-194	Human	↓PR	Impaired	[112]
miR-200/a	Human, mouse	↓ZEB1, PTEN, PGR, Bcl2	Variable between subtypes	[87,107,113,114]
miR-542	Human	↓IGFBP-1	Impaired	[115]

**Table 4 ijms-23-06210-t004:** miRNAs identified as potential serum biomarkers.

miRNA	Species	Target Gene/Protein	Improved/Impaired Receptivity	Ref.
miR-25	Human	n/a	Impaired	[141]
miR-27a	Human	↓IGF1	Impaired	[88]
miR-31	Human	↓FOXP3, CXCL12	Improved	[139]
miR-93	Human	n/a	Impaired	[141]
miR-106b	Human	n/a	Impaired	[141]
miR-146a	Human	↓STAT1	Improved	[141]
miR-152	Human	n/a	Impaired	[140]
miR-155	Human	↓SOCS1	Variable between papers	[140,141]

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
