# Peer review of "MicroRNAs in the Regulation of Endometrial Receptivity for Embryo Implantation"

_ijms, 2022, doi:10.3390/ijms23116210_

Round 1

Reviewer 1 Report

Authors summarized the anatomy of the human and murine uterus, the physiological changes during the human menstrual cycle and the murine estrous cycle, and major differences in embryo implantation and endometrial receptivity, basic molecular characteristics and biogenesis of miRNA and the role of miRNA in endometrial receptivity focused only on the literature evidences that were reported for human and mouse.

Concept and the structure of the review is very promising and well-planned, Abstract and Introduction catch the attention.

  1. Introduction flashes a promise of an extensive comparison of the miRNAs participated in the regulation of human and murine endometrial receptivity reflected to differences of anatomy and physiology.

Unfortunately, the scientific content of the continuing subsections, especially considering the “2. Endometrial cycle in the human and mouse”, “3.1 Overview of miRNAs” and “3.2 Overview of miRNAs and the endometrium” are represent general scientific outreaches with superficial comments on the literature findings. These parts do not meet the level of a scientific journal and need major revision together with the supporting figures.

 3.3. miRNAs and endometrial receptivity and 4. Potential clinical applications:  These are the most comprehensive parts of the paper, that could be abbreviated and submitted as a mini-review alone. Although these parts of the manuscript are eligible the connecting overviews are disintegrating in the text’s intelligibility and scientific quality.

  1. Conclusion is similar to the general overview parts, contain superficial, sweeping summary to the content

Figures 1-4 supporting these parts of the manuscript demonstrate overblown anatomical and physiological approaches, fitting more to a textbook. All figures were created with BioRender, widely used in scientific publications, that would allow detailed and complex representation and even comparison of the anatomical and histological differences in mouse and human.

Tables 1-4 are well edited and contain remarkable information content, supporting the understanding of the corresponding parts of the texts.

References: although the reference list is very extended and well-selected, I lack the level of the comparative interpretation of accumulated scientific data and author’s original aspect of the interpreted results.

Language editing of the manuscript is excellent.

Scientific novelty can only be found in subsection 3.3. miRNAs and endometrial receptivity and section 4. Potential clinical applications and in Table 1-4 in the recent form of the work. These parts are concise and eligible.

In summary, the review is a shallow synopsis of the current knowledge on the role of miRNA in endometrial receptivity I recommend to reject the manuscript in the current form and suggest resubmission after thorough revision.

Suggestions for major changes:

  1. Enhance the scientific information content and its display in the “2. Endometrial cycle in the human and mouse” section and “3.1 Overview of miRNAs” and “3.2 Overview of miRNAs and the endometrium” Deeper understanding and resolution of the physiologic changes can appear in the text and in the figures. These can be presented in fewer (1-2) complex figures rather than more simplified independent ones.
  2. Several publications present the miRNA findings according to early and mid-secretory phases (pre receptive and receptive endometrium) accurately timing the expression changes of the miRNA according to the LH peak:

Altmäe, Signe et al. “Meta-signature of human endometrial receptivity: a meta-analysis and validation study of transcriptomic biomarkers.” Scientific reports vol. 7,1 10077. 30 Aug. 2017, doi:10.1038/s41598-017-10098-3.

Liang J, Wang S, Wang Z. Role of microRNAs in embryo implantation. Reprod Biol Endocrinol. 2017 Nov 21;15(1):90. doi: 10.1186/s12958-017-0309-7. PMID: 29162091; PMCID: PMC5699189.

Similarly, in murine experiments, the miRNA changes can be related to the estrus cycle:

Chen K, Chen X, He J, et al. Mouse Endometrium Temporal and Spatial Expression mRNA and MicroRNA Associated With Embryo Implantation. Reproductive Sciences. 2015;22(11):1399-1408. doi:10.1177/1933719115580996

All of the aforementioned publications are already on the reference list of the manuscript, with numbers respectively: 3, 121 and 106. I suggest to use them for the basis of a deeper comparison.

  1. Subsection 1 Overview of miRNAs is completely unnecessary, a repetitively found in articles on miRNA from the 90’s and should be radically reduced or even omitted. 3.2. Overview of miRNAs and the endometrium can be a more interesting lead-up for the miRNA subsection including more elaborated interpretation of Reference No. 80 from the reference list.
  2. Conclusion section needs to include at least some specific comments on the differences and also on similarities of the role of miRNAs, that are conserved across the mammalian species and serve similar regulatory function (let-7 family, or PTEN and Wnt-signal regulator miRNAs).
  3. I suggest to keep the main structure of the manuscript, rather than abbreviate it- because an embracing mouse-human comparison on reproductive anatomy, physiology and molecular regulatory potentially evokes interest from wide audience of the research and clinical expert community.

Reviewer 2 Report

The manuscript by Shekibi et al, entitled “MicroRNAs in the regulation of endometrial receptivity for embryo implantation", is a very interesting review focused in shedding light into the molecular regulations underpinning endometrial transformation to a receptive state. This knowledge if fundamental to improving implantation rates in fertility treatments such as IVF.

The manuscript is well written, however, some changes could improve the quality of this review.

  • Chapter 2: Endometrial cycle in the human and mouse can be less descriptive since the main topic of the article is the molecular point of view. Specially the 2.2, where the differences between mouse uterus and estrous cycle with the human are out of context related to the main focus of the article.
  • Chapter 3.3.4 need to be further looked in depth. The topic relating contribution of the embryo-released molecules to endometrial receptivity signaling cascade is propelling to the forefront of investigations during the last years. The author only mentions in the last two sentences the role of embryonic mi-RNA in endometrial receptivity but more details are needed. I advice the author to check and cite these articles:
    • Dai Y, Qiu Z, Diao Z, Shen L, Xue P, Sun H, Hu Y. MicroRNA-155 inhibits proliferation and migration of human extravillous trophoblast derived HTR-8/SVneo cells via down-regulating cyclin D1. Placenta. 2012 Oct;33(10):824-9. doi: 10.1016/j.placenta.2012.07.012. Epub 2012 Aug 1. PMID: 22858023.
    • Cuman C, Van Sinderen M, Gantier MP, Rainczuk K, Sorby K, Rombauts L, Osianlis T, Dimitriadis E. Human Blastocyst Secreted microRNA Regulate Endometrial Epithelial Cell Adhesion. EBioMedicine. 2015 Sep 11;2(10):1528-35. doi: 10.1016/j.ebiom.2015.09.003. PMID: 26629549; PMCID: PMC4634783.
    • Xie L, Mouillet JF, Chu T, Parks WT, Sadovsky E, Knöfler M, Sadovsky Y. C19MC microRNAs regulate the migration of human trophoblasts. Endocrinology. 2014 Dec;155(12):4975-85. doi: 10.1210/en.2014-1501. Epub 2014 Sep 11. PMID: 25211593; PMCID: PMC4239420.

  • The next chapter, 3.3.5 is also extremely summarized and doesn’t present any specific information to the reader regarding the role of EV-s on endometrial receptivity. This topic can be further dived in by either explaining the role of specific miRNA found as part of the content of EV-s or the distinct miRNA profiling of EVs released by different biological sources such as uterine flushes, cultured endometrial epithelial cells etc. I advice the author to check and cite/discuss these articles for more in depth information regarding the role of EVs and their content in the regulation of endometrial receptivity:
    • Mishra A, Ashary N, Sharma R, Modi D. Extracellular vesicles in embryo implantation and disorders of the endometrium. Am J Reprod Immunol. 2021 Feb;85(2):e13360. doi: 10.1111/aji.13360. Epub 2020 Nov 4. PMID: 33064348.
    • Godakumara K, Ord J, Lättekivi F, Dissanayake K, Viil J, Boggavarapu NR, Faridani OR, Jääger K, Velthut-Meikas A, Jaakma Ü, Salumets A, Fazeli A. Trophoblast derived extracellular vesicles specifically alter the transcriptome of endometrial cells and may constitute a critical component of embryo-maternal communication. Reprod Biol Endocrinol. 2021 Jul 21;19(1):115. doi: 10.1186/s12958-021-00801-5. PMID: 34289864; PMCID: PMC8293585.
    • Governini L, Luongo FP, Haxhiu A, Piomboni P, Luddi A. Main actors behind the endometrial receptivity and successful implantation. Tissue Cell. 2021 Dec;73:101656. doi: 10.1016/j.tice.2021.101656. Epub 2021 Sep 29. PMID: 34634636
    • Jiang NX, Li XL. The Complicated Effects of Extracellular Vesicles and Their Cargos on Embryo Implantation. Front Endocrinol (Lausanne). 2021 Jun 4;12:681266. doi: 10.3389/fendo.2021.681266. PMID: 34149619; PMCID: PMC8213030.
    • Chen K, Liang J, Qin T, Zhang Y, Chen X, Wang Z. The Role of Extracellular Vesicles in Embryo Implantation. Front Endocrinol (Lausanne). 2022 Jan 28;13:809596. doi: 10.3389/fendo.2022.809596. PMID: 35154016; PMCID: PMC8831238
  • There are a lot of abbreviations for the target genes/proteins from the Table 1, Table 2 and Table 3 for which the full name of the gene/protein is missing throughout the art

Round 2

Reviewer 1 Report

The revised version takes into account all critical comments of both reviewers. I recommend to accept the paper in its present form.